# Lactobacilli and Their Probiotic Effects in the Vagina of Reproductive Age Women

**DOI:** 10.3390/microorganisms11030636

**Published:** 2023-03-01

**Authors:** Sonal Pendharkar, Axel Skafte-Holm, Gizem Simsek, Thor Haahr

**Affiliations:** 1Uvisa Health ApS, Boltonvej 21 A, 2300 Copenhagen, Denmark; 2Research Unit for Reproductive Microbiology, Department of Bacteria, Parasites and Fungi, Statens Serum Institut, 2300 Copenhagen, Denmark; 3Department of Biology, University of Copenhagen, 2200 Copenhagen, Denmark; 4Department of Gynecology and Obstetrics, Aarhus University Hospital, 8200 Aarhus, Denmark

**Keywords:** lactobacillus, vaginal microbiota, probiotics

## Abstract

In the present narrative review, the probiotic effects of vaginal *Lactobacillus* spp. are described in detail, covering the importance of the differential production of lactic acid, the lactic acid D/L isoforms, the questionable in vivo effect of hydrogen peroxide, as well as bacteriocins and other core proteins produced by vaginal *Lactobacillus* spp. Moreover, the microbe–host interaction is explained with emphasis on the vaginal mucosa. To understand the crucial role of *Lactobacillus* spp. dominance in the vaginal microbiota, different dysbiotic states of the vagina are explained including bacterial vaginosis and aerobic vaginitis. Finally, this review takes on the therapeutic aspect of live lactobacilli in the context of bacterial vaginosis. Until recently, there was very low-quality evidence to suggest that any probiotic might aid in reducing vaginal infections or dysbiosis. Therefore, clinical usage or over the counter usage of probiotics was not recommended. However, recent progress has been made, moving from probiotics that are typically regulated as food supplements to so-called live biotherapeutic products that are regulated as medical drugs. Thus, recently, a phase 2b trial using a *Lactobacillus crispatus* strain as a therapeutic add-on to standard metronidazole showed significant reduction in the recurrence of bacterial vaginosis by 12 weeks compared to placebo. This may constitute evidence for a brighter future where the therapeutic use of lactobacilli can be harnessed to improve women’s health.

## 1. Introduction

*Lactobacillus* species *(spp.)* are part of the normal human microbiota found in the mouth, gastrointestinal tract, and cervicovaginal tract. The discovery of a distinct vaginal microbiota was made in 1891 by gynaecologist Albert Döderlein, who was the first to report evidence of what is now known as *Lactobacillus* spp. [1]. Today, it is well-established that a symbiotic relationship exists between reproductive age women and their vaginal *Lactobacillus* spp. The vaginal lactobacilli feed on glycogen byproducts of oestrogenised vaginal epithelial cells [2] and, in turn, the lactobacilli produce lactic acid to create a low-pH environment favouring their own survival over opportunistic pathogens [3,4,5]. In 2011, the hallmark study conducted by Ravel and colleagues used the 16S rRNA marker gene to establish the so-called community state types (CSTs), stratifying the vaginal microbiota into four different low-diversity CSTs dominated by only a single *Lactobacillus (L.)* spp.: *L. crispatus* (CST-I), *L. gasseri* (CST-II), *L. iners* (CST-III) and *L. jensenii* (CST-V)*,* respectively [6]. Roughly 80% of reproductive age women have a simplistic *Lactobacillus* spp. dominant vaginal microbiota, whereas the remaining 20% harbour a more diverse community state type (CST-IV) that has multiple sub-groups [7]. When women are not dominated by lactobacilli, it has been shown that they have increased susceptibility to many genital infections, including *Chlamydia trachomatis,* herpes and HIV [8,9,10]. Moreover, women not dominated by lactobacilli may be at increased risk of many adverse reproductive outcomes, e.g., poor fecundity rate (FR = 0.57) [11], poor pregnancy rate (RR = 0.55) [12], early spontaneous abortion (RR = 1.68) [13], late miscarriage (OR = 6.32), preterm delivery (OR = 2.38) and maternal infection (OR = 2.53) [14]. Hence, intervention-based studies with lactobacilli as therapeutic agents are increasingly being conducted aiming to recover *Lactobacillus* dominance in the genital tract microbiota, which is then perceived to provide a better long-term cure against infections and to improve reproductive health. Probiotics are typically regulated as food supplements and regarded as safe [15]; importantly, however, they are not medical drugs. Thus, due to the poor regulation with food supplements, there is poor clinical evidence as well as limited safety data available on probiotics. A new frontier is emerging with so-called live biotherapeutic drugs that contain live microorganisms and need to be tested as classical medical drugs. In this narrative review, we aim to provide an up-to-date extraction of the current evidence on vaginal lactobacilli, their physiological role in balancing eubiosis and dysbiosis, the interplay with the host, and their therapeutic effect against bacterial vaginosis as reported in clinical trials.

## 2. Physiological Role of *Lactobacillus* spp. in Maintaining Eubiosis in a Healthy Vagina

As mentioned, only a few *Lactobacillus* spp. predominate the healthy vaginal microbiota including *L. crispatus, L. jensenii, L. gasseri, L. iners*, with each species having its own specific defence mechanism. See Figure 1 for an overview of *Lactobacillus* and non-lactobacillus related defence mechanisms. In a variety of ways, different *Lactobacillus* spp. and strains exhibit probiotic effects and outcompete other bacteria, and protect against reproductive tract pathogens such as *Chlamydia trachomatis,* HSV-2 and HIV-1 [16,17,18].

### 2.1. Vaginal Lactobacillus spp. and Lactic Acid

The vaginal microbiota is influenced by oestrogen, causing changes in the composition of the microbiota throughout a woman’s lifespan. During puberty, oestrogen levels increase and support the availability of glycogen byproducts in the vaginal epithelium, which are nutritionally essential for *Lactobacillus* spp. and form the foundation for the mutualism between the woman’s cervicovaginal tract and the microbiota [19]. Among reproductive-age women, a high abundance of lactobacilli is a hallmark of a healthy environment [20]. The *Lactobacillus* spp.-dominant vagina has a mean pH of around 3.80 ± 0.20, which is predominantly caused by lactic acid secreted from the lactobacilli [3]. This relates to the fermentation process of glucose, where the break-down products of glycogen are utilised under anaerobic conditions in the production of lactic acid, which lowers the vaginal pH [6]. This acidification serves as a physiological defence mechanism to inhibit the growth of pathogenic bacteria by permeating cell membranes and inducing osmotic stress [5], by causing destabilisation of the outer membrane in Gram-negative bacteria, and potentially enhance the effect of other immunomodulatory and antimicrobial properties [21]. Although women are typically dominated by one or the other *Lactobacillus* spp., it may be that a consortia of different *Lactobacillus* spp. acting in symbiosis is important to provide maximum protection against invading pathogens [22].

In a comparison study, all the studied *L. gasseri* and *L. crispatus* (including the CTV-05 strain which is present in the live biotherapeutic product LACTIN-V) strains killed *Prevotella bivia* CI-1 and *Gardnerella* sp. 594, pyelonephritis-associated *Escherichia coli* CFT073, and recurrent cystitis- and preterm labour-associated IH11128 *E. coli* through direct contact [23]. This antimicrobial activity was associated with increased lactic acid secretion. *L. crispatus* and *L. iners* have different fermentation pathways, through which *L. crispatus* can metabolise both glucose and lactose. *L. crispatus* has lactate dehydrogenase (LDH) for both the isomers of lactate, thus producing both D and L forms of lactate. In contrast, *L. iners* metabolises glucose and has LDH that is specific to producing the L-lactate isoform. This is especially interesting because of the limited protection provided by *L. iners* against pathogens which may potentially be related to the inferior L-lactate isoform [24]. Similar observations were also made when the four predominant vaginal *Lactobacillus* spp. were compared for their lactic acid production, the two isoforms, and their influence on the host gene expression [20]. The D isoforms of lactic acid produced by *L. crispatus*, *L. jensenii* and *L. gasseri* were negatively correlated with the levels of vaginal extracellular matrix metalloproteinase-inducer (EMMPRIN) and matrix metalloproteinase-8 (MMP-8), which are known to alter the tight junctions in the endocervical epithelium, making the female genital tract susceptible to infection. In the vaginal discharge of women dominated by *L. iners* or *Gardnerella*, the L- isoform of lactic acid was significantly higher, making the L-D lactic acid ratio skewed towards the L-isoform. This was positively correlated to the significant increase in EMPRRIN and MMP-8 levels, which might explain why vaginal microbiota dominated by *L. iners* is more susceptible to dysbiosis and associated complications. This association was also established in relation to different vaginal disorders with similar outcomes in a Brazilian study, where L-lactic acid was significantly associated with increased levels of EMPRRIN and MMP-8 [25]. Another important aspect of lactic acid and acidity was carefully studied by O’Hanlon et al. [3,26], who reported that the protonated form of lactic acid (LAH) is microbicidal and is the result of LAH, the lactate anion (LA^-^) and hydrogen ion (H^+^) concentration (pH). They suggest a more accurate measure of acidity in vaginal microbiota dominated by lactobacilli by keeping strict inclusion criteria, maintaining hypoxic condition (as in the vagina) throughout the sample collection and processing, and correcting for CO_2_ loss.

### 2.2. Lactobacilli and Hydrogen Peroxide

The production of hydrogen peroxide (H_2_O_2_) by *Lactobacillus* spp. seems important in maintaining a healthy vaginal microbiota. In general, studies have ascribed H_2_O_2_ a protective role against BV [27,28] and sexually transmitted infections [28] due to the presence of H_2_O_2_-producing *Lactobacillus* spp. In vitro experiments have shown that isolates of *L. crispatus*, *L. jensenii*, and *L. gasseri* were superior to *L. iners* in the number of isolates producing H_2_O_2_ [29]. The role in vivo, however, remains speculative as the facultative anaerobic environment of the vagina emphasises the implausible explanation of this causality. The amount of oxygen (O_2_) required to produce H_2_O_2_ is limited in the cervicovaginal tract due to the hypoxic environment, which makes it questionable whether the physiological conditions allow contribution of this protective property [30]. More likely, the antimicrobial properties are attributed to other characteristics including lactic acid, the ability to compete with other microorganisms for available resources in the vagina [31] and to competitively prevent infection/colonisation of unfavourable microorganisms by inhibiting their binding to the epithelium [19]. Moreover, some *Lactobacillus* spp. produce antimicrobial proteinaceous substances, i.e., bacteriocins, to neutralise closely related species through bactericidal or bacteriostatic activity [32].

### 2.3. Lactobacilli, Bacteriocins and Core Proteins

Molecular studies on lactobacilli have deciphered some mechanisms of host–microbe or microbe–microbe interactions [33,34]. These mechanisms include production of antimicrobial peptides (bacteriocins), S-layer proteins and adhesion-associated compounds. Ojala et. al. studied the genomes of 10 different *L. crispatus* strains and found genes encoding collagen-binding adhesins, exopolysaccharides (EPS), and bacteriocins, such as bacteriolysin and pediocin-like bacteriocins in vaginal isolates, which, taken together, can explain how *L. crispatus* strains prevent adherence of *Gardnerella* spp. to the epithelial cells [35]. Some other bacteriocins from vaginal lactobacilli such as Lactocin 160 from *L. rhamnosus* [36] and HV219 from *Lactococcus lactis* subsp. HV219 [37] were found to be active against *Gardnerella* sp. by disrupting the cell membrane and other Gram-variable bacteria. Recently, both *L. crispatus* and *L. gasseri* strains were shown to produce emulsifying biosurfactants which were active against candida by reducing their adherence to epithelial cells [38,39]. Although specifically effective in inhibiting pathogens, bacteriocins are not widely studied and are much overlooked.

### 2.4. L. iners and Its Differential Role

A very interesting *Lactobacillus* sp. that has been a riddle is *L. iners*. It is part of the normal vaginal microbiota and, in fact, it is one of the predominant vaginal species; however, its role in vaginal health is ambiguous. *L. iners* has an unusually smaller genome when compared to other lactobacilli in the vagina, which is indicative of a parasitic and symbiotic lifestyle [40]. It is also now known that a specific cholesterol-dependent cytolysin (CDC), “inerolycin”, is produced by *L. iners,* which is a pore-forming toxin akin to vaginolysin of *Gardnerella* sp. [40,41]. It can hence be hypothesised that the presence of *L. iners* as a predominant coloniser offers a favourable environment for *Gardnerellla* sp. to survive and destabilise the vaginal microbiota predominantly due to the limited amount of lactate produced. Increasing evidence suggests that *L. iners* provides less protection against infectious species and is more prone to bacterial vaginosis acquisition [24], compared to the other *Lactobacillus* spp., which could be explained by its physiological- and biochemical limitations.

## 3. Natural Non-Lactobacillus-Based Defence Mechanisms of the Vagina

Both the microbiota and host play crucial roles in the protection against pathogens. The vaginal mucosa is the host’s first line of defence and serves as a physical, chemical, and immunological barrier against potential exogenous pathogens [27]. The natural vaginal defence mechanism can be divided into three crucial components: the vaginal immune system, the stratified squamous vaginal epithelium (VE), and the mucus layer. Vaginal immunity presents to the host upon the occurrence of pathogenic species, whereas the VE and mucus layer are the surface upon which pathogenic species adhere to initiate infections [28,29]. The VE layers are held together through tight junction proteins, which help maintain the cellular integrity of the VE layers and limit the dissemination of pathogens [30]. The VE surface expresses several Toll-like receptors (TLR) [31]. These receptors recognize pathogen-associated molecular patterns (PAMPS), mediate the release of antimicrobial peptides (AMP), and initiate other innate immune responses. AMPs target pathogens via direct killing and immune modulation through the recruitment and activation of immune cells [32,33]. Defensins is a class of AMPs also secreted in the vagina that has multiple mechanisms of action by directly killing and inhibiting bacterial toxins against pathogens, including HIV and HPV [34,35]. Another important component of the host defence mechanism is surfactants, which are pattern-recognition molecules of the collectin family of C-type lectin. The surfactant protein A is located in the deep epithelial layers and in the superficial epithelial layer [36]. They are a part of innate immune response and act primarily through the opsonization of pathogens to support phagocytosis, as well as by modulating the adaptive immune response by interacting with antigen-presenting cells (APCs). Meanwhile, another surfactant protein D showed inhibitory effects preventing HIV-1’s passage through the EpiVaginal tissue compartments in vitro, indicating its role in epithelial barrier function [37].

The VE is coated by mucus that provides a dense lubricated barrier. Mucus is mainly composed of water, ions, lipids, and highly glycosylated glycoproteins (mucins) [42]. The VE expression of mucin genes was observed to be lower in the vagina than in the cervix, suggesting mucus is predominantly secreted from the cervix, where the mucus functions by way of entrapment of the ascending microbes [43]. Another component of mucus is immunoglobulin (Ig). The vaginal mucus predominantly contains IgG [44]. IgG prevents pathogenic species from adhering to the host cell by contributing to the opsonization of pathogenic species. However, the complete defence mechanism is not fully understood [45,46]. Furthermore, mucus plays an important role in regulating the VE interplay with microbiota. It is suggested that mucin supports bacterial adhesion of beneficial microbes. Binding protein domains that allow adhesion to mucins have been found on commensal species, including *L. crispatus* and *L. gasseri* [47]. Moreover, the superficial VE layer undergoes rapid regeneration throughout a women’s natural cycle. The exfoliation of VE cells serves as a protective function by disengaging exogenous pathogens from direct contact with the woman [48]. APCs in the VE and mucus layer are crucial in the role of recognizing and modulating immune responses to various antigens from exogenous pathogens. The effective immune response depends on the APCs’ ability to process antigens and present them to T cells to activate the immune response [49]. The APCs in the female genital tract include dendritic cells (DCs), macrophages, and Langerhans cells (LCs) [50,51]. LCs, a specialised subset of DCs, reside within the VE and are the predominant APC. Following contact with antigen LC emigrates from VE to draining lymph nodes, LC presents the antigen to T lymphocytes, and initiates an immune response. Studies have shown that LC plays a crucial role in virus dissemination, including HIV [47,52].

## 4. Vaginal Dysbiosis: Understanding the Abnormal Microbial Conditions of the Vagina

The vaginal microbiota of reproductive-age women represents a delicate balance that is challenged frequently during the menstrual cycle [53], e.g., by internal factors such as reproductive hormones and menstruation but also by external factors such as douching [54] and having multiple sexual partners [55]. In the presence of one or more of the above-mentioned challenges, the vaginal microbiota might shift from a *Lactobacillus* spp.-dominant to a *Lactobacillus* spp.-deficient state with increasing bacterial diversity including either strict anaerobes such as *Prevotella, Mobiluncus, Gardnerella* sp., *Atopobium vaginae* (now *Fannyhessia vaginae*) [56], or aerobic species. These two common bacterial dysbiotic states are known as bacterial vaginosis (BV) and aerobic vaginitis (AV) and both of them can be identified through microscopy. BV is a common dysbiosis characterized by reduced or no lactobacilli with the strict anaerobes mentioned above in dominance [57,58]. In contrast, AV is a condition characterized by reduced lactobacilli and overgrowth of *Escherichia coli, Streptococcus* spp. (including *group B streptococci*), *Enterococcus faecalis, Staphylococcus epidermidis*, and/or coagulase-negative *Staphylococci* spp. [59,60]. BV and AV are distinct with a significant difference in the inflammatory response, with AV presenting a classical inflammatory response [61], whereas BV can be more or less silent/sub-clinical [62]. In 95–100% of cases of BV, *Gardnerella* sp. is identified as the most likely causative agent [63,64]. Research suggests that *Gardnerella*’s pathogenicity is dependent on its virulence factors, such as sialidase activity, and its toxin vaginolysin, and that the species and strain level diversity is high among *Gardnerella* spp., making some species/strains more dependable on other BV-associated bacteria for causing infection [65,66,67]. Biofilm formation through *Gardnerella* sp. is also key in establishing infection as it tolerates acidity and H_2_O_2_, protecting *Gardnerella* sp. while assisting other BV-associated bacteria including *Prevotella bivia* to grow and form a polymicrobial biofilm [68]. This biofilm also provides antimicrobial resistance and promotes recurrent or persistent infection/dysbiosis [69].

It is increasingly evident that a vaginal microbiota low in *Lactobacillus* spp. and high in diversity—usually CST IV—is associated with increased incidence of sexually transmitted infections (STI), including the risk of HIV and HPV acquisition [55,56]. Interestingly, the spontaneous clearance of dysplasia from HPV infection is associated with low-diversity *Lactobacillus* spp. dominant vaginal microbiota, whereas disease progression is significantly more often observed in high-diversity communities such as some CST III and most CST IV [70,71].

In 1994, Benson J. Horowitz and colleagues reported on a new condition in women with profuse vaginal discharge. Through microscopy [72], plenty of unusually long lactobacilli-related species (average length of 60 µm—named leptothrix) were found. In comparison, *Lactobacillus* spp. in healthy women have a length between 5–15 µm. The syndrome was named lactobacillosis [73]. The cause of this morphologic alteration remains unexplained, but has been linked to diabetes mellitus, misuse of antifungal medications, and vulvodynia [72]. Vaginal lactobacillosis can be treated with antibiotics [73]. Another closely related condition is cytolytic vaginosis, which has the typical symptoms of white cheesy vaginal discharge, pruritus, and vulvar dysuria. Cytolytic vaginosis also has an overgrowth of *Lactobacillus* spp., is referred to as ‘supernormal flora’ due to the excessive amount of lactobacilli [74] and is diagnosed via wet smear displaying numerous lactobacilli, low pH, paucity of white blood cells, evidence of cytolysis and absence of *Trichomonas*, *Gardnerella*, and *Candida* sp. [75]. Sodium bicarbonate douching can be used as a treatment to increase the vaginal pH [75].

To sum up, vaginal dysbiosis is not a single entity and the indigenous lactobacilli are either absent, lowered or present in abnormally high numbers, see Figure 2.

## 5. Vaginal Probiotic Supplements and Live Biotherapeutic Drugs: Success and Challenges

The Nobel Prize winner Elie Metchnikoff (1845–1916) was the first to hypothesise the health benefits of consuming live microorganisms, which was later conceptualised in medicine, to improve human health [76]. In 1998, Andrew W. Bruce and Gregor Reid first reported that the therapeutic use of lactobacilli can confer protection against pathogenic species. In this first clinical study, intravaginal administration of *Lactobacillus casei* GR-1 prevented the emergence of coliform bacteria and cured women for a period ranging from four weeks to six months [77]. The World Health Organization defines probiotics as ‘*live microorganisms that, when administered in adequate amounts, confer a health benefit on the host*’ [78]. Today, with the growing body of evidence from microbiome-based research, a new frontier has emerged for evidence-based research and development of therapeutic microorganisms intended for clinical usage. The FDA in the US now refers to live biotherapeutic products which “(*1) contains live organisms, such as bacteria; (2) is applicable to the prevention, treatment, or cure of a disease or condition of human beings; and (3) is not a vaccine*.” Thus, according to the FDA, it is essential to evaluate the efficacy and safety in phase 1, 2 and 3 trials—just as any other drug—before making claims for a certain clinical condition.

Much of the probiotic research has been conducted aiming to treat BV. BV has traditionally been treated with metronidazole and antibiotics; however, BV is often recurrent or treatment-resistant. Moreover, the use of antibiotics has side effects and carries a risk of antibiotic resistance which altogether necessitates new treatment strategies. Thus, as a means to reduce the above-mentioned drawbacks of antibiotics, research has been conducted with therapeutic administration of probiotics containing lactobacilli to restore a healthy vaginal microbiota. Most commonly, probiotics intended to recover vaginal health are administered orally or vaginally. Despite gastrointestinal transit after oral intake, lactobacilli have proved to survive the acidic milieu in the upper gastrointestinal tract [79]. Administration of *L. rhamnosus* (GR-1) and *L. fermentum* (RC-14) has been shown to reduce urogenital infections in women [80]; however, RCT-studies on the same strains/product in relation to pregnancy have shown low [81] or no modifying effects on the vaginal microbiota [82]. Hence, the use of at least the above-mentioned strains orally, and perhaps the oral usage of probiotics generally, is somewhat controversial when it comes to improving vaginal health.

In an early study in 2006, the effect of probiotics in BV treatment was reported to an 88% recovery rate in Nigerian women treated vaginally with capsules containing *L. rhamnosus* (GR-1) and *L. reuteri* (RC-14) and noted fewer side effects and better cure rates compared to vaginal metronidazole-gel [83]. Later in 2009, a double-blind, placebo-controlled clinical trial including 39 Italian women with BV revealed that vaginally administered tablets containing *L. brevis* (CD2), *L. salivarius* (FV2), and *L. plantarum* (FV9) had a curative effect. In contrast to the placebo-treated group who had a 12% recovery rate, a total of 83% were BV-free after 7-days of active treatment [84]. A systematic review and meta-analysis from 2009 reported – based on one small study - that vaginal gelatine tablets containing lactobacilli were more effective than oral metronidazole in the treatment of BV (RR 0.20, 95% CI 0.05–0.80) [85]. Later in 2013, a meta-analysis showed that probiotics in combination with antibiotics significantly improved the cure rate of BV (RR = 1.53, 95% CI 1.19–1.97), although a large heterogeneity was observed between studies [86]. In 2020, a meta-analysis found an even more pronounced effect on the recovery of normal vaginal microbiota after 1 month when treated with probiotics in combination with antibiotics for BV as compared to antibiotics plus placebo probiotics (OR = 4.55, 95% CI: 1.44–14.36) [87]. In support, one of the most recent meta-analyses published on the topic in 2021 [88], it was shown that optimal clinical cure rate of BV was achieved with antibiotics administered either locally or orally when combined with probiotics delivered intravaginally as compared to the average treatment of all other treatment modalities.

The long-term effects of vaginal probiotics are not thoroughly examined. One study by Larsson PG et. al. using long-term follow-up for 24 months found that the recurrence of BV was reduced in women treated with both metronidazole and clindamycin, followed by vaginal capsules containing different strains of lactobacilli [89]. The same group further investigated the effect of combining vaginal probiotics following an aggressive antimicrobial treatment in women suffering from BV and recurrent vulvovaginal candidiasis [90]. Women were followed up until 12 months post treatment and the use of probiotics increased the cure rate to 89% at 12 months for vulvovaginal candidiasis and to 67% for BV. To our knowledge, the only vaginal live biotherapeutic product is LACTIN-V, which consists of *L. crispatus* CTV-05. LACTIN-V has been proven effective as an add-on to vaginal metronidazole treating BV in a recent phase-2b trial [91]. In this trial, women (*n* = 228) testing positive for BV were allocated in a 2:1 ratio to repeated doses of LACTIN-V for 11 weeks (*n* = 152) versus placebo (*n* = 76), all receiving vaginal metronidazole for 5 days prior to randomization. The primary outcome was the recurrence of BV by week 12 after the treatment. The LACTIN-V arm provided a significant reduction in recurrent BV, RR = 0.66, 95% CI 0.44–0.87.

In addition to probiotics, postbiotics have attracted researchers to study and test them for the treatment of biofilms formed by pathogens [92]. Postbiotics are bioactive compounds or metabolites that are produced by probiotic strains and examples are lactic acid [93], extracellular vesicles (EVs), and metabolites that help maintain the healthy homeostasis of the vagina. Biosurfactants and bacteriocins, as discussed earlier, are important mediators of a healthy vaginal environment. Focus is also being drawn to the studying of the symbiotic effects of combining probiotics with pre and postbiotics, which is promising as it helps stabilise vaginal pH, and introduce beneficial lactobacilli to balance the microbiome, and the post biotics help maintain low pH, decrease pathogen adhesion, and inhibit quorum sensing to disrupt biofilm formation, preventing the recurrence of infection [92].

Despite some success with probiotic studies, the clinical application of probiotics is very limited and without proper evidence to support systematic usage. As discussed above, orally administered probiotics have survived the acid milieu of the upper gastrointestinal tract; however, their success is not consistent throughout the studies and the mode of administration is one questionable factor among others. Carefully selected intravaginal probiotic strains are expected to show higher colonisation compared to those administered orally simply because of the higher and adequate availability of the administered strains on site. Cure rates that are registered immediately after the treatment with antimicrobials alone or together with probiotics are expected to be higher than those registered several weeks post cessation of the treatment because of the influencing physiological and lifestyle factors. The contraceptive method used and the relationship status of the participants will also have an influence on their BV status and respective cure rates, which should be considered while assessing the success of the studies [89]. Therefore, large and properly designed studies with a suitable route of probiotic administration for maximum availability of the probiotic strains, appropriately combined with antimicrobials and the participants being followed-up for a longer duration (at least three months), will create the much needed and awaited evidence for therapeutic potential of vaginal probiotics.

## 6. Regulatory Processes and Changes Required for Making Advancements

Probiotics are typically regulated as food supplements and not as medical drugs; thus, the poor clinical evidence available on probiotics as well as limited safety data are typically due to poor regulation. For this reason, it was not possible to make recommendations for a specific probiotic product in a recent systematic review on vaginal probiotics [94]. Apart from probiotics most often being regulated as food supplements, it is worrisome that the European CE marking for medical devices actually has allowed probiotics and prebiotics to be classified as medical devices class IIa, e.g., EcovagBalance^®^ vaginal capsule, which is commercially available as an over-the-counter product in Denmark [95]. The worrisome aspect in this regulation is that while CE marking allows the product/device to make efficacy claims, there is no legal requirement to actually perform clinical studies supporting these claims [96]. Therefore, although “*marketers need to ensure that they hold robust evidence for their medical claims (Rule 12.1) and are reminded that whilst CE certification may demonstrate that the device is safe and fit for its intended purpose, a CE certification in itself does not constitute evidence for the purposes of the rule*” [97]. This seems to be a loophole in the regulation, effectively allowing the marketing of products with medical claims without securing proper evidence to support them. In a recent commentary, it was speculated that poor regulation might effectively prevent good quality research and, thus, the advancement of science and clinical practice in the probiotic field as large RCTs are increasingly demanding and expensive and may turn out to be negative or neutral, showing no effect on the intervention [98]. The recent LACTIN-V trial [46] shows that, fortunately, this type of research is still being conducted.

## 7. Future Perspectives on Vaginal Lactobacilli and Their Probiotic Usage

In a world where antimicrobial resistance is declared one of the biggest threats to global health [99], alternative treatments and strategies to lower antibiotic usage are necessary. Probiotics represent one alternative. The new frontier with probiotics being studied properly as medical drugs to show efficacy and safety before clinical usage seems to pave the way for a brighter future. Until recently, there was no evidence to support any probiotic use to improve vaginal microbiota [47]. However, the live biotherapeutic product LACTIN-V showed great efficacy in reducing recurrent BV as an adjuvant to vaginal metronidazole in a phase-2b trial. To the best of our knowledge LACTIN-V remains the only live biotherapeutic product intended for vaginal use undergoing clinical studies. Obviously, other probiotic *Lactobacillus* strains deserve further study, either for individual usage or possibly together in a *Lactobacillus* “cocktail” product. Other novel interventions include pre and postbiotics, bacteriophages, and vaginal microbiota transplantation, which was recently investigated in five patients suffering from recurrent bacterial vaginosis [100]. Although the treatment effect from vaginal microbiota transplantation appeared promising, the potential confounding effects of co-intervention with vaginal antibiotics were not clearly identified in the study. Further studies are warranted.

The optimal dosage of *Lactobacillus* products also needs further study. It seems that many probiotic intervention-based studies administer a dose around 10^8^–10^9^ colony-forming units based on the physiological vaginal *Lactobacillus* concentration, but only a limited number of dose-finding studies are available. It could be speculated whether a higher dose and several doses a day would enable higher colonisation rates. Another issue is the timing of probiotic administration—should it be alongside antibiotic therapy or immediately after antibiotic therapy? At bedtime or any time? During menstruation or after menstruation? Additionally, although partner treatment is not recommended by the latest Cochrane review on BV [101], there seems to be a biological rationale for partner treatment.

## Figures and Tables

**Figure 1 microorganisms-11-00636-f001:**
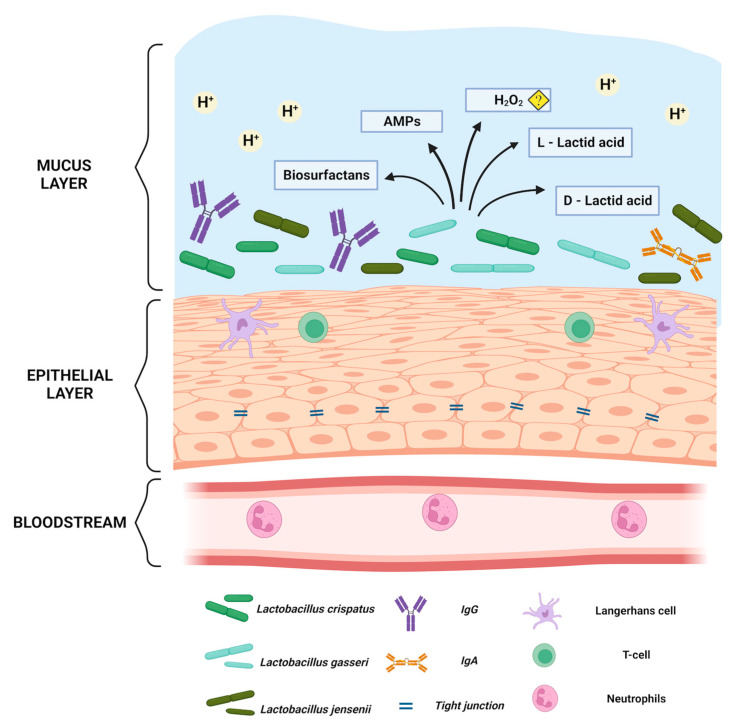
The normal *Lactobacillus* spp. dominant vaginal mucosa. Created with BioRender.com. Illustration showing below the bloodstream under the basal membrane, from where neutrophils can be attracted if pathogens enter the vagina. The vaginal squamous epithelium is connected through tight junctions which inhibit pathogens to enter basal epithelial layers. The epithelium also has antigen-presenting cells as well as t-cells. In the mucus, there is an acidic pH ≤ 4 predominantly due to the lactic acid produced by the vaginal lactobacilli. In addition, there are numerous defence mechanisms including antimicrobial peptides (AMPs) and biosurfactants. The role of hydrogen peroxide is questionable *in vivo* as it depends on the existence of oxygen.

**Figure 2 microorganisms-11-00636-f002:**
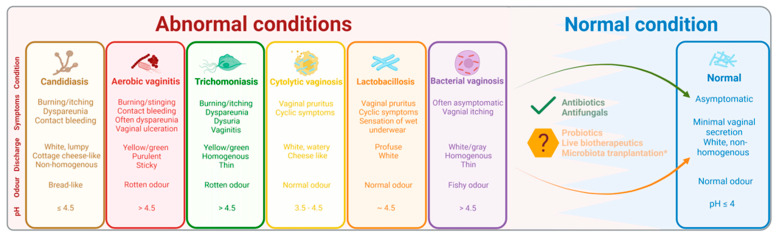
The normal and most common abnormal microbial conditions of the vagina. Created with BioRender.com. Schematic figure showing the characteristics of the most common abnormal microbial conditions of the vagina. Proper diagnosis and treatment can lead to a normal condition. However, recurrent infection and treatment resistance/failure is more and more common, paving the way for new treatment modalities such as probiotics, live biotherapeutics and microbiota transplantation.

## Data Availability

Not applicable.

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
