# Peer review of "Lactobacilli and Their Probiotic Effects in the Vagina of Reproductive Age Women"

_microorganisms, 2023, doi:10.3390/microorganisms11030636_

Round 1

Reviewer 1 Report

Manuscript ID Microorganisms-2165190

Reviewer report

Congratulations to the authors for the present review on lactobacilli and their probiotic effects in the vaginal microbiota. The present work is very interesting for the scientific community worldwide and deserves to be published without any doubt.

The manuscript is well-written, and I recommended some improvements that should beneficiate the manuscript. However, most of the comments or recommendations are not mandatory and only should be followed if the author considers them useful.

The remaining observations are type errors that I found during my reading of the manuscript.

All suggestions are summarized in my minor comments. Please check below.

Minor comments

Abstract

Lines 12, 14, and 15- Please put “spp.” in non-italics form. Also, check the same error in the remaining manuscript.

Line 13- Please put “in-vivo” in italics form.

Lines 15 and 16- Please add the comma after “To understand the crucial role of Lactobacillus spp. dominance in the vaginal microbiota” and replace “is” with “are”.

Main text of the manuscript

Line 98- Please add sp. or species after “Gardnerella”. Also, check the same error in the remaining manuscript.

Line 117- Please maintain the same term “In vitro” or “In-vivo” in italics form, as written in the line 119. Also, check the same error in the remaining manuscript.

Line 141- Please clarify “Gram-variant bacteria”.. Do you mean Gram-variable bacteria?

Line 146- Please replace “Lactobacillus spp.” with “Lactobacillus sp.”.

Line 154-157- “Increasing evidence suggests that L. iners provides less protection against infectious species and is more prone to bacterial vaginosis acquisition [20], compared to the other Lactobacillus spp. and could be explained by its physiological- and biochemical limitations.”

It is true that L. iners provides less protection by itself and its genome size is lower that other lactobacilli. However, it is important to clarify that although different Lactobacillus species may play very different role in vaginal microbiota, the consortia formed by different lactobacilli is equally important for a healthy vaginal protection. Please check:

https://pubmed.ncbi.nlm.nih.gov/35646732/  or https://doi.org/10.3389/fcimb.2022.863208

Pacha-Herrera D, Erazo-Garcia MP, Cueva DF, Orellana M, Borja-Serrano P, Arboleda C, Tejera E, Machado A. Clustering Analysis of the Multi-Microbial Consortium by Lactobacillus Species Against Vaginal Dysbiosis Among Ecuadorian Women. Front Cell Infect Microbiol. 2022 May 11;12:863208. doi: 10.3389/fcimb.2022.863208.

I invite the authors to address this topic in order to avoid misunderstanding of the Readers. Perhaps in a subsection with one or two paragraphs. I do believe that it is worth mentioning that the probiotic activity is caused not only by individual Lactobacillus species but also by its multi-microbial interaction. However, the probiotic activity promoted by multi-microbial consortia is still unknown. This recommendation is not mandatory, but I do believe that it will improve the authors’ point of view if the authors consider it relevant.

Page 4 and 5 on subsection “3. Natural non-lactobacillus-based defence mechanisms of the vagina”- I recommend the authors to also address the surfactant proteins role as an important innate immune preventive response being an effective prophylactic strategy in inhibiting vaginal infection or dysbiosis. There are several papers in the literature, for example:

https://pubmed.ncbi.nlm.nih.gov/14678203/

MacNeill C, Umstead TM, Phelps DS, Lin Z, Floros J, Shearer DA, Weisz J. Surfactant protein A, an innate immune factor, is expressed in the vaginal mucosa and is present in vaginal lavage fluid. Immunology. 2004 Jan;111(1):91-9. doi: 10.1111/j.1365-2567.2004.01782.x. PMID: 14678203; PMCID: PMC1782386.

Or

https://pubmed.ncbi.nlm.nih.gov/30984160/

Pandit H, Kale K, Yamamoto H, Thakur G, Rokade S, Chakraborty P, Vasudevan M, Kishore U, Madan T, Fichorova RN. Surfactant Protein D Reverses the Gene Signature of Transepithelial HIV-1 Passage and Restricts the Viral Transfer Across the Vaginal Barrier. Front Immunol. 2019 Mar 28;10:264. doi: 10.3389/fimmu.2019.00264. PMID: 30984160; PMCID: PMC6447669.

Lines 208 and 209 – Please rectify “Atopobium (now Fannyhessia vagina)” with “Atopobium vaginae (now Fannyhessia vaginae)”. Fannyhessea vaginae is the current species for the species previously known as Atopobium vaginae, as you may consult in:

https://www.ncbi.nlm.nih.gov/Taxonomy/Browser/wwwtax.cgi?mode=Info&id=82135

Lines 235-236- Please replace “unusually long lactobacilli-related spp.” with “unusually long lactobacilli-related species”.

Line 246- Please add species after “Trichomonas, Gardnerella, and Candida”.

Lines 271-276- “Much of the probiotic research has been conducted aiming to treat BV. BV has traditionally been treated with metronidazole and other antibiotics, however, BV is often recurrent or treatment resistant. Moreover, the use of antibiotics has side effects and carries risk of resistance which altogether necessitates new treatment strategies. Thus, as a means to lower the above mentioned drawbacks of antibiotics, research has been conducted with therapeutic administration of probiotics containing lactobacilli to restore a healthy vaginal ecosystem.”

I recommend the following reference and the authors to check the following meta-analysis on the evaluation of antibiotics and probiotics effectiveness by itself and combined for BV treatment:

Muñoz-Barreno, A.; Cabezas-Mera, F.; Tejera, E.; Machado, A. Comparative Effectiveness of Treatments for Bacterial Vaginosis: A Network Meta-Analysis. Antibiotics 202110, 978. https://doi.org/10.3390/antibiotics10080978

Line 371- Please also include postbiotics and synbiotics as additional examples of other interventions.

Again, congratulations to the authors for the present study. It was an honor to read your review on this topic.

Author Response

Dear Reviewer 1,

We thank you for reviewing our manuscript and sharing detailed observations with us. We identify your comments, suggestions and greatly appreciate you for directing us towards supporting literature. Below we are addressing your comments & suggestions:

Minor comments

Lines 12, 14, and 15- Please put “spp.” in non-italics form. Also, check the same error in the remaining manuscript.

Response: We have now put “spp.” in non-italics in Lines 12, 14, 15 and in the rest of the manuscript.

Line 13- Please put “in-vivo” in italics form.

Response: We have now put “in-vivo” in italics in Line 13 and have checked the rest of the manuscript.

Lines 15 and 16- Please add the comma after “To understand the crucial role of Lactobacillus spp. dominance in the vaginal microbiota” and replace “is” with “are”.

Response: We have now added a comma after “To understand the crucial role of Lactobacillus spp. dominance in the vaginal microbiota” in Line 16 and replaced “is” with “are” in Line 17.

Main text of the manuscript

Line 98- Please add sp. or species after “Gardnerella”. Also, check the same error in the remaining manuscript.

Response: Thank you. We have now added sp. or spp. after “Gardnerella” where appropriate and have checked/rectified the same in the rest of the manuscript.

Line 117- Please maintain the same term “In vitro” or “In-vivo” in italics form, as written in the line 119. Also, check the same error in the remaining manuscript.

Response: We have now checked and aligned the text to write “In vitro” or “In-vivo” in italics in Line 140 and in the rest of the document.

Line 141- Please clarify “Gram-variant bacteria”.. Do you mean Gram-variable bacteria?

Response: Yes, we mean Gram-variable bacteria and have corrected “Gram-variant bacteria” to Gram-variable bacteria.

Line 146- Please replace “Lactobacillus spp.” with “Lactobacillus sp.”.

Response: We have now replaced “Lactobacillus spp.” with “Lactobacillus sp.” and checked the same in the rest of the document.

Line 154-157- “Increasing evidence suggests that L. iners provides less protection against infectious species and is more prone to bacterial vaginosis acquisition [20], compared to the other Lactobacillus spp. and could be explained by its physiological- and biochemical limitations.”

It is true that L. iners provides less protection by itself and its genome size is lower that other lactobacilli. However, it is important to clarify that although different Lactobacillus species may play very different role in vaginal microbiota, the consortia formed by different lactobacilli is equally important for a healthy vaginal protection. Please check:

https://pubmed.ncbi.nlm.nih.gov/35646732/  or https://doi.org/10.3389/fcimb.2022.863208

Pacha-Herrera D, Erazo-Garcia MP, Cueva DF, Orellana M, Borja-Serrano P, Arboleda C, Tejera E, Machado A. Clustering Analysis of the Multi-Microbial Consortium by Lactobacillus Species Against Vaginal Dysbiosis Among Ecuadorian Women. Front Cell Infect Microbiol. 2022 May 11;12:863208. doi: 10.3389/fcimb.2022.863208.

I invite the authors to address this topic in order to avoid misunderstanding of the Readers. Perhaps in a subsection with one or two paragraphs. I do believe that it is worth mentioning that the probiotic activity is caused not only by individual Lactobacillus species but also by its multi-microbial interaction. However, the probiotic activity promoted by multi-microbial consortia is still unknown. This recommendation is not mandatory, but I do believe that it will improve the authors’ point of view if the authors consider it relevant.

Response: Thank you for this invitation. We agree and have added this aspect in line 88-90.

Page 4 and 5 on subsection,3. Natural non-lactobacillus-based defence mechanisms of the vagina”- I recommend the authors to also address the surfactant proteins role as an important innate immune preventive response being an effective prophylactic strategy in inhibiting vaginal infection or dysbiosis. There are several papers in the literature, for example:

https://pubmed.ncbi.nlm.nih.gov/14678203/

MacNeill C, Umstead TM, Phelps DS, Lin Z, Floros J, Shearer DA, Weisz J. Surfactant protein A, an innate immune factor, is expressed in the vaginal mucosa and is present in vaginal lavage fluid. Immunology. 2004 Jan;111(1):91-9. doi: 10.1111/j.1365-2567.2004.01782.x. PMID: 14678203; PMCID: PMC1782386.

Or

https://pubmed.ncbi.nlm.nih.gov/30984160/

Pandit H, Kale K, Yamamoto H, Thakur G, Rokade S, Chakraborty P, Vasudevan M, Kishore U, Madan T, Fichorova RN. Surfactant Protein D Reverses the Gene Signature of Transepithelial HIV-1 Passage and Restricts the Viral Transfer Across the Vaginal Barrier. Front Immunol. 2019 Mar 28;10:264. doi: 10.3389/fimmu.2019.00264. PMID: 30984160; PMCID: PMC6447669.

Response: We agree with the comment on discussing the role of surfactant proteins in innate immunity against vaginal infection and dysbiosis. As suggested, we have now included a comment on surfactants with relevant references between Lines 167-169 and 205-213.

Lines 208 and 209 – Please rectify “Atopobium (now Fannyhessia vagina)” with “Atopobium vaginae (now Fannyhessia vaginae)”. Fannyhessea vaginae is the current species for the species previously known as Atopobium vaginae, as you may consult in:

https://www.ncbi.nlm.nih.gov/Taxonomy/Browser/wwwtax.cgi?mode=Info&id=82135

Response: We have now corrected “Atopobium” to “Atopobium vaginae” in Lines 246-247.

Lines 235-236- Please replace “unusually long lactobacilli-related spp.” with “unusually long lactobacilli-related species”.

Response: We have now replaced “unusually long lactobacilli-related spp.” with “unusually long lactobacilli-related species”.

Line 246- Please add species after “Trichomonas, Gardnerella, and Candida”.

Response: We have now added “sp.” after “Trichomonas, Gardnerella, and Candida”.

Lines 271-276- “Much of the probiotic research has been conducted aiming to treat BV. BV has traditionally been treated with metronidazole and other antibiotics, however, BV is often recurrent or treatment resistant. Moreover, the use of antibiotics has side effects and carries risk of resistance which altogether necessitates new treatment strategies. Thus, as a means to lower the above mentioned drawbacks of antibiotics, research has been conducted with therapeutic administration of probiotics containing lactobacilli to restore a healthy vaginal ecosystem.”

I recommend the following reference and the authors to check the following meta-analysis on the evaluation of antibiotics and probiotics effectiveness by itself and combined for BV treatment:

Muñoz-Barreno, A.; Cabezas-Mera, F.; Tejera, E.; Machado, A. Comparative Effectiveness of Treatments for Bacterial Vaginosis: A Network Meta-Analysis. Antibiotics 2021, 10, 978. https://doi.org/10.3390/antibiotics10080978

Response: As suggested we have now added the abovementoned meta-analysis in Lines “342 to 345.

Line 371- Please also include postbiotics and synbiotics as additional examples of other interventions.

Response: We have now added our comment on postbiotics and synbiotics from Lines 364-374.

Reviewer 2 Report

The paper by Kulkarni et al describes a review of the probiotic effects in the vagina of reproductive-age women of Lactobacilli. This is an interesting topic, particularly because very recently probiotics for vaginal application have very recently fell within the scope of the Medicinal Products Directive (MPD). However, this is not reflected in the text. In the same line of thought, the authors have not extensively described probiotics for vaginal application that have already been described - a search in pubmed yields 600 results. The authors need to update their literature. Small details that need improvement are listed below:

line 152: I think here the authors mean "L. iners" instead of "inerolycin"

A review needs tables and/or figures.

Author Response

Dear Reviewer 2,

We thank you for reviewing our manuscript and sharing your feedback with us. We have considered your feedback and have tried to incorporate as much as possible in the given time frame. Please find our responses to your comments & suggestions:

  1. The paper by Kulkarni et al describes a review of the probiotic effects in the vagina of reproductive-age women of Lactobacilli. This is an interesting topic, particularly because very recently probiotics for vaginal application have very recently fell within the scope of the Medicinal Products Directive (MPD). However, this is not reflected in the text. In the same line of thought, the authors have not extensively described probiotics for vaginal application that have already been described - a search in pubmed yields 600 results. The authors need to update their literature.

Response: We disagree about the postulate that the text does not reflect the fact that probiotics for vaginal application now fall within the scope of MPD. Actually, this is mentioned already in the abstract Line 22-26 and discussed thoroughly in the paragraph “Regulatory processes and changes required for making advancement “. Although this is not a systematic review, we agree to the need of discussing other probiotic trials in our paper as you also suggested and thank you for pointing out the same. Addressing this feedback, we have now added more probiotics trials and meta-analyses that are relevant for this review. This addition can be found from Lines 334-349.

Minor: Small details that need improvement are listed below:

  1. line 152: I think here the authors mean "L. iners" instead of "inerolycin".

Response: We have now corrected “inerolycin” to “L. iners”.

  1. A review needs tables and/or figures.

Reponse: As suggested, we have now created two figures.

Round 2

Reviewer 2 Report

The authors made a great effort reviewing the manuscript, which has been much improved. The Figures look especially good. Great work!